# Estimating long-term vaccine effectiveness against SARS-CoV-2 variants: a model-based approach

Alexandra B. Hogan[1,2], Patrick Doohan [2], Sean L. Wu[3], Daniela Olivera Mesa [2], Jaspreet Toor [2], Oliver J. Watson [2,4], Peter Winskill [2], Giovanni Charles[2], Gregory Barnsley [2,4], Eleanor M. Riley[5], David S. Khoury [6], Neil M. Ferguson [2] & Azra C. Ghani [2] ✉

With the ongoing evolution of the SARS-CoV-2 virus updated vaccines may be needed. We fitted a model linking immunity levels and protection to vaccine effectiveness data from England for three vaccines (Oxford/AstraZeneca AZD1222, Pfizer-BioNTech BNT162b2, Moderna mRNA-1273) and two variants (Delta, Omicron). Our model reproduces the observed sustained protection against hospitalisation and death from the Omicron variant over the first six months following dose 3 with the ancestral vaccines but projects a gradual waning to moderate protection after 1 year. Switching the fourth dose to a variant-matched vaccine against Omicron BA.1/2 is projected to prevent nearly twice as many hospitalisations and deaths over a 1-year period compared to administering the ancestral vaccine. This result is sensitive to the degree to which immunogenicity data can be used to predict vaccine effectiveness and uncertainty regarding the impact that infection-induced immunity (not captured here) may play in modifying future vaccine effectiveness.

The rapid development and roll-out of SARS-CoV-2 vaccines has had a major effect on the health impacts of the global pandemic, substantially reducing COVID-19 cases, hospitalisations, and deaths[1–3]. Despite several vaccines showing high initial efficacy against infection with the ancestral virus, the sequential emergence of variants of concern has substantially reduced the effectiveness of vaccines in blocking infection and onward transmission, although efficacy against severe outcomes has been more durable[4–6]. The emergence and global spread of the Omicron variant and its subtypes has resulted in repeated infection due to waning and reduced effectiveness of vaccine- and infection-induced immunity[7,8]. Omicron has now replaced prior variants globally and has been the dominant variant circulating for over 1 year, albeit with several emerging sub-variants[9]. Two Omicron-specific bivalent vaccines, that include antigens representing both the

ancestral Wuhan virus and Omicron subtypes, are now available[10]. These have demonstrated higher immunogenicity against the Omicron BA.1 subvariant and against the BA.4/BA.5 subvariants than the ancestral vaccines[11,12].

As SARS-CoV-2 continues to evolve, it is likely that both existing and updated variant-specific vaccines will lag viral antigenic evolution. Moreover, as is currently the case for influenza vaccines, decisions regarding investment in, or introduction of, new vaccines, as well as assessment of the need for further boosting, will likely be based on immunogenicity and safety data rather than clinical trials. Obtaining reliable data on vaccine efficacy will be hampered by the high degree of infection-induced, broad-based antiviral immunity among most of the world's population, making the identification of appropriate comparator groups challenging. This ongoing interaction between

[1]School of Population Health, Faculty of Medicine and Health, University of New South Wales, Sydney, NSW, Australia. [2]MRC Centre for Global Infectious Disease Analysis, School of Public Health, Imperial College London, London, UK. [3]Institute for Health Metrics and Evaluation, University of Washington, Seattle, USA. [4]London School of Hygiene and Tropical Medicine, London, UK. [5]Institute of Immunology and Infection Research, School of Biological Sciences, University of Edinburgh, Edinburgh, UK. [6]Kirby Institute, University of New South Wales, Sydney, NSW, Australia. ✉e-mail: a.ghani@imperial.ac.uk

infection-induced immunity and vaccination (so-called "hybrid immunity"[13]) will also influence the effectiveness of vaccine booster programmes.

A method for estimating vaccine efficacy from immunogenicity data was proposed by Khoury et al.[14], who demonstrated that neutralising antibody titres (NATs) could act as a correlate of protection across a range of SARS-CoV-2 vaccines. In this, a non-linear dose–response model is estimated to relate NAT to protection against different clinical endpoints—capturing the more rapid decline in protection against mild disease that occurs as NAT declines over time compared to the slower decline in protection against more severe endpoints. Using this model, they subsequently predicted the loss of efficacy against emerging variants[15], as well as more recently the potential benefit of introducing variant-specific vaccines against a range of circulating variants[16]. One of the limitations with a model based on NAT alone is that it does not capture the broader immune responses generated by vaccination (or infection) and how this may differ between vaccines[17]. For example, studies have demonstrated that the adenovirus-vectored AstraZeneca AZD1222 vaccine induced broad T-cell responses even though the level of NAT induced following vaccination is lower than that of the mRNA vaccines[18]. It is also suggested that inactivated whole virus vaccines (such as the VLA2001 vaccine manufactured by Valneva) should induce even more broadly

based immune responses, which are in turn predicted to be both more durable and less susceptible to viral immune escape[19].

Here we infer a simple model of immune waning and boosting directly from clinical endpoints. Using a similar model framework to that developed by Khoury et al.[14] we infer the underlying immune dynamics by fitting to national-level vaccine effectiveness estimates from England. Although the immunological mechanisms of protection from severe disease are not entirely clear, NAT are a well-established mechanistic correlate of protection from infection. Whilst cellular immunity is likely to play an additional role in protection from severe COVID-19, for simplicity we assume that a single immunological marker providing a surrogate for protection against infection can capture patterns of protection from severe disease. In so doing, we obtain estimates of vaccine effectiveness against three endpoints—symptomatic mild disease, hospitalisation, and death—for combinations of three widely used vaccines. By incorporating follow-up through 2021 and 2022, we estimate the vaccine effectiveness against both the Delta variant (dominant in 2021) and the Omicron BA.1/BA.2 variants (circulating in the first half of 2022). We use the fitted model to provide short-term projections of vaccine effectiveness, noting the considerable uncertainty associated with these, and to estimate the potential effectiveness of variant-adapted vaccines.

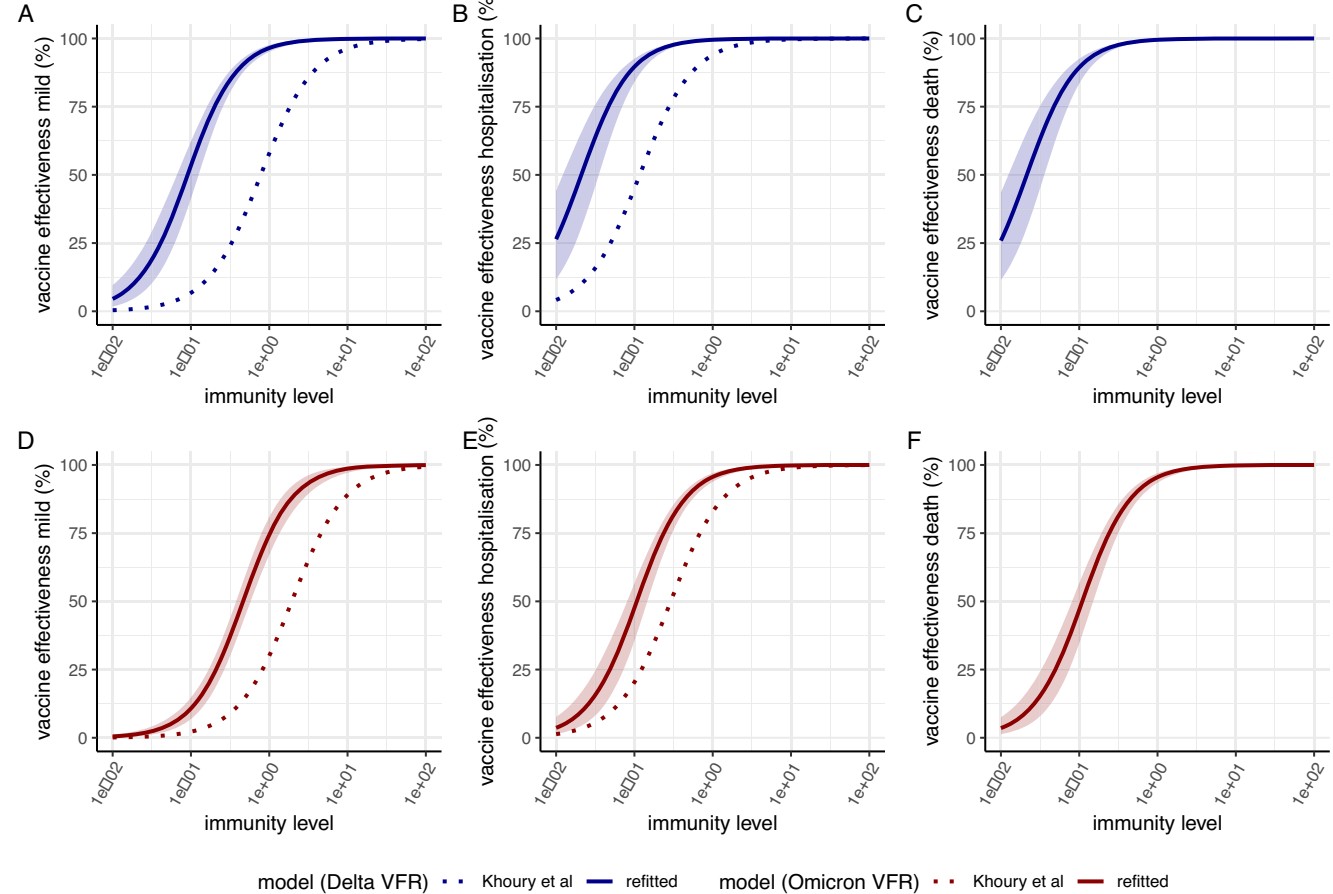

**Fig. 1 | Relationship between immunity levels and vaccine effectiveness.** Dose–response curves estimated from fitting to vaccine effectiveness data for the relationship between immunity level (IL, x-axis) and vaccine effectiveness against mild disease (**A**, **D**), hospitalisation (**B**, **E**) and death (**C**, **F**). Panels (**A**–**C**) show vaccine effectiveness against the Delta variant whilst panels (**D**–**F**) show vaccine effectiveness against the Omicron/BA.1 variant. The solid lines show the posterior median estimates and colour bands the 95% credible interval from our model fitted to the data on vaccine effectiveness against each variant. The dotted lines show the dose–response curves that would be projected using the original efficacy model presented in Khoury et al.[14] which fitted the relationship between NAT and clinical endpoints against the ancestral virus and adjusting for the Delta and Omicron variants respectively by using the variant fold reductions (VFRs) from immunogenicity data reported in Cromer et al. and Khoury et al.[15,16].

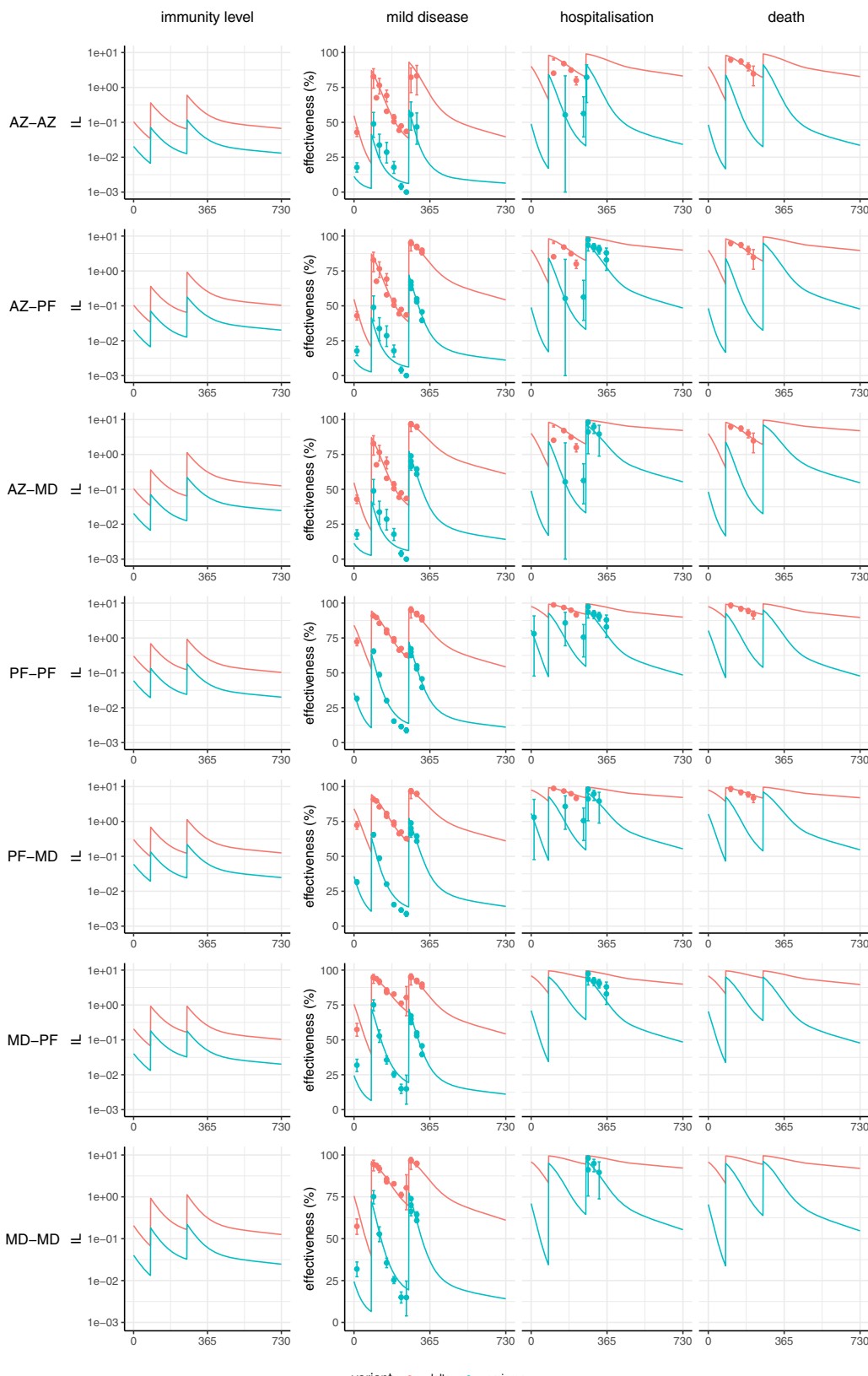

## Results

Figure 1 shows our estimated relationship between immune level and protection against mild disease, hospitalisation and death for the Delta and Omicron variants obtained by fitting the model to data on vaccine effectiveness over time. The shape of these curves is consistent with the observation that higher immune levels are required for protection against mild disease than for protection against the more severe endpoints (hospitalisation and death). This model generates a good fit to the observed vaccine effectiveness data for the 3 vaccines delivered in England and reproduces the differential rates of decline in vaccine effectiveness observed against both the Delta and Omicron variants over 1 year of follow-

**Fig. 2 | Projected vaccine effectiveness over time (in days) since the first vaccine dose for combinations of schedules for the Oxford/AstraZeneca AZD1222 (AZ), Pfizer-BioNTech BNT162b2 (PF) and Moderna mRNA-1273 (MD) vaccines.** Plots show immunity level (IL) in the left column, alongside effectiveness against mild disease, hospitalisation, and death on the right. Neutralisation and protection against the Delta and Omicron variants are shown in red and blue respectively. Seven regimens are shown: AZ delivered for three doses (AZ–AZ); two doses of AZ and a third dose of PF (AZ–PF); two doses of AZ and a third dose of MD (AZ–MD); PF

delivered for three doses (PF-PF); two doses of PF and a third dose of MD (PF-MD); two doses of MD and a third dose of PF (MD–PF); and MD delivered for three doses (MD–MD). The solid lines show the posterior median fitted model estimate, and the points show mean estimates of vaccine effectiveness (and associated 95% confidence intervals) against three endpoints using data from England[23–25]. Where estimates were only available stratified by age group, the data are from the 65+ age group. Sample sizes for each of the 143 data points vary according to the uptake of combinations and length of follow-up; these are provided in the data files.

## Table 1 | Prior and posterior parameter estimates for the immunological model

| Parameter | Symbol | Prior mean (95% confidence interval) | Posterior median (95% credible interval) | Reference for prior mean |
|---|---|---|---|---|
| **Immunity levels for each vaccine based on prior data on NAT** | | | | |
| *Oxford/AstraZeneca AZD1222 vaccine* | | | | |
| IL against Delta for dose 1 relative to convalescent | $n_{AZ,1}$ | 0.07 (0.005, 1.04) | 0.10 (0.08, 0.14) | - |
| IL against Delta for dose 2 relative to convalescent | $n_{AZ,2}$ | 0.14 (0.09, 0.22) | 0.36 (0.29, 0.45) | 14,15,35 |
| IL against Delta for dose 3 relative to convalescent | $n_{AZ,3}$ | 2.0 (0.13, 30.27) | 0.60 (0.45, 0.82) | 30,36 |
| *Pfizer-BioNTech BNT162b2 vaccine* | | | | |
| IL against Delta for dose 1 relative to convalescent | $n_{PF,1}$ | 0.30 (0.02, 4.56) | 0.30 (0.24, 0.38) | - |
| IL against Delta for dose 2 relative to convalescent | $n_{PF,2}$ | 0.61 (0.39, 0.96) | 0.69 (0.55, 0.87) | 14,15,37 |
| IL against Delta for dose 3 relative to convalescent | $n_{PF,3}$ | 2.0 (0.13, 30.27) | 0.92 (0.71, 1.19) | 30,36 |
| *Moderna mRNA-1273 vaccine* | | | | |
| IL against Delta for dose 1 relative to convalescent | $n_{MD,1}$ | 0.53 (0.04, 7.96) | 0.20 (0.16, 0.27) | - |
| IL against Delta for dose 2 relative to convalescent | $n_{MD,2}$ | 1.06 (0.68, 1.67) | 0.93 (0.72, 1.19) | 14,38 |
| IL against Delta for dose 3 relative to convalescent | $n_{MD,3}$ | 2.0 (0.13, 30.27) | 1.13 (0.86, 1.45) | 30,36 |
| **Immune escape parameters** | | | | |
| Fold-reduction for Omicron relative to Delta (all vaccines) | VFR | 1.0 (0.001, 2.96) | 5.1 (4.0, 6.8) | 39 |
| **Immunity level decay parameters** | | | | |
| Half-life of IL decay: short (days) | $h_s$ | 58 (48, 68) | 35 (31, 38) | 14,39 |
| Half-life of IL decay: long (days) | $h_l$ | 270 (-122, 662) | 581 (354, 872) | 40 |
| Time period for switching (days) | $t_s$ | 90 (31, 149) | 75 (60, 88) | 14,39 |
| **Relationship between IL and protection** | | | | |
| IL relative to convalescent required to provide 50% protection from mild disease | $n_{50_1}$ | 0.2 (0.13, 0.31) | 0.091 (0.066, 0.125) | Prior mean and 95% confidence interval[14] |
| IL relative to convalescent required to provide 50% protection from hospitalisation | $n_{50_2}$ | 0.03 (0.01, 0.13) | 0.021 (0.012, 0.035) | Prior mean and 95% confidence interval[14] |
| IL relative to convalescent required to provide 50% protection from death | $n_{50_3}$ | 0.03 (0.01, 0.13) | 0.021 (0.012, 0.036) | Prior based on hospitalisation[14] |
| Shape parameter | $k$ | 2.94 (1.76, 4.12) | 3.2 (2.7, 3.7) | Prior mean and 95% confidence interval[14] |

up (Figs. 2 and S1). We estimate a 5.1-fold (95% CrI 4.0–6.8) reduction in IL (induced by vaccination with the ancestral Wuhan strain of the virus) against the Omicron variant relative to the Delta variant.

We compared this relationship to that expected if the original relationship between NAT and protection inferred from clinical trial data against the Wuhan virus by Khoury et al.[14] is adjusted based on immunogenicity data alone (i.e. without further model fitting to clinical data against new strains). The dashed lines in Fig. 1 show this predicted relationship for the Delta and Omicron viruses, respectively, by applying a 3.9-fold reduction from the Wuhan virus to Delta[15] and 9.7-fold reduction from the Wuhan virus to the Omicron BA.1 variant[16], respectively. For both the Delta and the Omicron variants, applying the fold reductions estimated from immunogenicity data to the relationship inferred against the Wuhan virus results in a more pessimistic prediction of vaccine effectiveness than was inferred from fitting directly to vaccine effectiveness data against the Delta and Omicron variants.

We additionally explored the sensitivity of our results to different model structures. The alternative model in which protection against hospitalisation and death is conditional on protection against infection (as a constant scaling) provided the best overall fit to the data used in the fitting (Table S3). The relationship between immunity levels and protection against severe disease and death for this model differs from the main model, with less of a reduction in predicted vaccine effectiveness at lower immunity levels (Fig. S2). Thus, this model predicts more sustained protection over time (Fig. S2). However, we did not take this model forwards as it was inconsistent with longer-term follow-up data published in aggregate form which show a decline in vaccine effectiveness that better aligns with our main model (Table S4)[20]. The alternative boosting model produced a poorer fit to the data compared to our main model (Table S3) but generated a similar relationship between immunity levels and protection (Fig. S3, Table S5).

The fitted model parameters are summarised in Table 1. In terms of comparative effectiveness of the three vaccines, we

estimate a trend with the highest IL generated with mRNA-1273 followed by BNT162b2 and then AZD1222, consistent with the empirical data which shows higher levels of peak protection following both dose 2 and dose 3. The half-life of IL during the initial more rapid period of decay is estimated to be 35 days (95% Credible Interval (CrI) 31–38 days), shorter than the estimate of 58 days estimated by Khoury et al.[14] for neutralisation titre decay following infection although our model structure includes a more gradual transition to the slow delay over a 75-day period. The estimate of the half-life for the subsequent longer period of decline of 581 days (95% CrI 354–872 days) was consistent with the 500 days assumed by Khoury et al.[14], and with the wide uncertainty bounds indicating a remaining degree of uncertainty in the longer-term durability of protection.

Table 2 shows estimates of vaccine effectiveness over time against the Omicron variant (comparative estimates against the Delta variant are given in Table S1 and using the 18–64 age data in Table S2). We distinguish the estimates obtained within the time period of the data observations from those beyond the data period (i.e. short-term projections), noting that the latter have additional uncertainty associated with the parametric assumptions made in our model that is not reflected in the confidence intervals. These values should therefore be interpreted with this uncertainty in mind.

Using these short-term projections, we estimate that 180 days after administering the third dose, effectiveness against hospitalisation with Omicron declines to 49.7% (95% CrI 42.2–56.5%) for AZD1222, 70.3% (95% CrI 68.0–71.8%) for mRNA-1273 and 64.1% (95% CrI 61.7–65.7%) for BNT162b2. These are in line with recent estimates of vaccine effectiveness (with a slightly different case definition, see methods) in England across all vaccines which show 65.3% vaccine effectiveness (95% CI 61.7–68.6%) between 3 and 6 months following the 3rd dose[20]. One year after vaccination, these levels are predicted to decline further, resulting in relatively low protection against infection or mild disease, and moderate protection against hospitalisation (38.0%, 95% CrI 29.8–45.3% for AZD1222; 59.5%, 95% CrI 53.9–62.7% for mRNA-1273; 52.6%, 95% CrI 46.9–56.0% for BNT162b2). This is also in line with the more recent data which show 51.1% vaccine effectiveness (95% CI: 45.7–56.0%) between 9 and 12 months following dose 3[20]. Fitting to the younger age-group results in similar estimates of vaccine effectiveness, albeit with a slightly steeper decline in vaccine effectiveness against hospitalisation and death (Table S2).

We used the estimates of the relative neutralisation titres of variant-adapted vaccines compared to ancestral vaccines reported by Khoury et al.[16] to explore the potential benefit of variant-adapted vaccines. They estimated a 1.61-fold (95% CI 1.5-1.8) increase in NAT against across all strains compared to the ancestral vaccines, with a higher increase (1.85, 95% CI 1.6–2.1) against homologous strains and lower increase (1.47, 95% CI 1.3–1.7) against heterologous strains. We applied this increase to the immunogenicity generated by the mRNA.1273 vaccine to illustrate the value of variant-matched vaccines (the equivalent tables for AZ1222 and BNT162b2 are provided in Supplementary Information). The resulting estimates of vaccine effectiveness are shown in Table 3. The estimated vaccine effectiveness curves against any strain, and separately against homologous and heterologous strains, following a 4th dose are shown in Fig. 3A. Compared with administration of the ancestral vaccine as a fourth dose, we estimate relatively little difference in levels of protection against hospitalisation and death shortly after the 4th dose is administered between the two vaccines, relative to the much more substantial impact that administering any fourth dose is predicted to have compared to not administering a fourth dose (Fig. 3B), consistent with the results in Khoury et al.[16]. However, this pattern is not sustained over time with a predicted more rapid drop in efficacy against both mild disease and hospitalisation

**Table 2 | Estimated vaccine effectiveness against mild disease, hospitalisation and death for Oxford/AstraZeneca AZD1222, Pfizer-BioNTech BNT162b2 and Moderna mRNA-1273 vaccine regimens as a function of time since dose 2 or dose 3**

| Vaccine | Days post dose 2 | | Days post dose 3 | | | | | | |
|---|---|---|---|---|---|---|---|---|---|
| | 90 | 180 | 30 | 60 | 90 | 120 | 150 | 180 | 365 |
| *Mild disease* | | | | | | | | | |
| Oxford/AstraZeneca AZD1222 | 12 (11.5–12.4) | 6.1 (5.8–6.4) | 42.4 (35.8–48.9) | 29.9 (24.4–35.7) | 21.6 (17.2–26.4) | 16.5 (13–20.4) | 13.4 (10.4–16.8) | 11.6 (8.9–14.5) | 7.5 (5.4–9.8) |
| Moderna mRNA-1273 | 33.3 (32.1–34.1) | 19.3 (18.1–20) | 63.8 (62.6–64.3) | 50.5 (49.1–51.4) | 39.7 (38.2–40.7) | 32.1 (30.6–33.1) | 27.1 (25.7–28.1) | 23.9 (22.5–24.9) | 16.3 (13.7–17.9) |
| Pfizer-BioNTech BNT162b2 | 24.9 (24.1–25.4) | 13.7 (12.9–14.1) | 57.1 (56.2–57.4) | 43.5 (42.5–44.1) | 33.2 (32.1–33.9) | 26.3 (25.3–27) | 22 (20.9–22.6) | 19.2 (18.1–19.9) | 12.9 (10.6–14.2) |
| *Hospitalisation* | | | | | | | | | |
| Oxford/AstraZeneca AZD1222 | 50.7 (48.9–52.1) | 32.9 (30.9–34.4) | 84.7 (80.7–88) | 76.2 (70.8–80.8) | 67.4 (61–73.1) | 59.8 (52.6–66.2) | 53.9 (46.6–60.6) | 49.7 (42.2–56.5) | 38 (29.8–45.3) |
| Moderna mRNA-1273 | 79 (77.5–80) | 64.2 (61.7–65.9) | 93 (92.4–93.3) | 88.5 (87.6–89) | 83.2 (81.9–84.1) | 78 (76.4–79.2) | 73.7 (71.7–75) | 70.3 (68–71.8) | 59.5 (53.9–62.7) |
| Pfizer-BioNTech BNT162b2 | 71.4 (69.9–72.5) | 54.3 (52–55.9) | 90.9 (90.3–91.3) | 85.3 (84.3–85.9) | 78.9 (77.6–79.8) | 72.9 (71.2–74) | 67.9 (65.8–69.2) | 64.1 (61.7–65.7) | 52.6 (46.9–56) |
| *Death* | | | | | | | | | |
| Oxford/AstraZeneca AZD1222 | 50 (46–53.8) | 32.3 (28.8–35.9) | 84.4 (79.3–87.8) | 75.8 (68.9–80.7) | 66.8 (58.9–72.9) | 59.1 (50.8–66) | 53.3 (44.6–60.5) | 49 (40.3–56.3) | 37.4 (28.8–44.8) |
| Moderna mRNA-1273 | 78.6 (75.5–81.1) | 63.6 (59.2–67) | 92.8 (91.6–93.7) | 88.2 (86.4–89.6) | 82.8 (80.3–84.9) | 77.6 (74.4–80.1) | 73.2 (69.5–75.9) | 69.7 (65.7–72.7) | 58.9 (52.3–63.6) |
| Pfizer-BioNTech BNT162b2 | 70.8 (67.3–73.9) | 53.7 (49.3–57.3) | 90.7 (89.1–91.8) | 85 (82.7–86.6) | 78.5 (75.5–80.8) | 72.4 (68.8–75.2) | 67.4 (63.4–70.5) | 63.5 (59.3–66.9) | 52 (45.4–56.6) |

Estimates are shown for the Omicron variant with the ancestral vaccines; estimates for the Delta variant are shown in Table S2. The values are shown for each vaccine for dose 2 or dose 3; as the dose 3 estimates do not depend on dose 2 values, the estimates are applicable to either homologous or heterologous dosing. Values shown are the posterior median and 95% credible intervals. Bold indicates projected vaccine effectiveness beyond the time period of the data to which the model was fitted.

**Table 3 | Projected vaccine effectiveness against mild disease, hospitalisation and death from BA.1 for an ancestral vaccine (mRNA.1273) and variant-adapted vaccine as a function of time since a fourth dose**

| Vaccine | Days post 4th dose | | | | | | |
|---|---|---|---|---|---|---|---|
| | 30 | 60 | 90 | 120 | 150 | 180 | 365 |
| *Mild disease* | | | | | | | |
| Moderna mRNA.1273 | 63.8 (62.6–64.3) | 50.5 (49.1–51.4) | 39.7 (38.2–40.7) | 32.1 (30.6–33.1) | 27.1 (25.7–28.1) | 23.9 (22.5–24.9) | 16.3 (13.7–17.9) |
| Variant-adapted vaccine | 78.9 (78–82.4) | 68.4 (67.2–73.1) | 58.2 (56.8–63.7) | 50 (48.6–55.7) | 44.1 (42.5–49.7) | 39.9 (38.3–45.4) | 29.3 (26.3–34.9) |
| *Hospitalisation* | | | | | | | |
| Moderna mRNA.1273 | 93 (92.4–93.3) | 88.5 (87.6–89) | 83.2 (81.9–84.1) | 78 (76.4–79.2) | 73.7 (71.7–75) | 70.3 (68–71.8) | 59.5 (53.9–62.7) |
| Variant-adapted vaccine | 96.6 (96.3–97.3) | 94.2 (93.8–95.4) | 91.3 (90.8–93) | 88.3 (87.5–90.6) | 85.6 (84.6–88.3) | 83.4 (82.2–86.5) | 75.7 (72.3–80.4) |
| *Death* | | | | | | | |
| Moderna mRNA.1273 | 92.8 (91.6–93.7) | 88.2 (86.4–89.6) | 82.8 (80.3–84.9) | 77.6 (74.4–80.1) | 73.2 (69.5–75.9) | 69.7 (65.7–72.7) | 58.9 (52.3–63.6) |
| Variant-adapted vaccine | 96.5 (96–97.4) | 94.1 (93.3–95.5) | 91.1 (90–93.2) | 88 (86.5–90.7) | 85.3 (83.5–88.6) | 83 (80.9–86.7) | 75.2 (71.4–80.8) |

Values shown are the posterior median and 95% credible intervals. The comparator group is those that did not receive any vaccine dose.

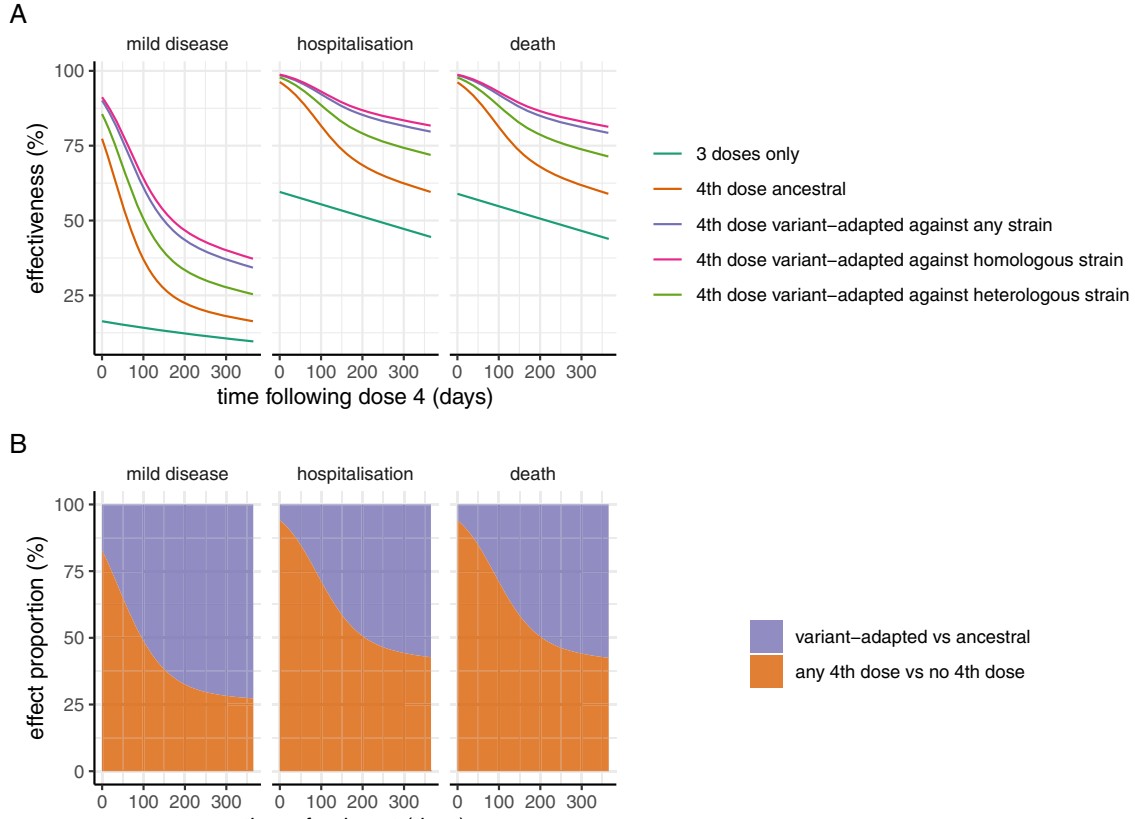

**Fig. 3 | Model projected vaccine effectiveness over time following a fourth dose with either the vaccine against the ancestral strain (illustrated with Moderna mRNA.1273) or a variant-adapted vaccine, compared to only three doses.** **A** Vaccine effectiveness against mild disease, hospitalisation, and death, for three doses only (turquoise line), a fourth dose with the ancestral Moderna vaccine (orange line), a fourth dose with a variant-adapted vaccine against any strain (purple line), a variant-adapted vaccine against a homologous strain (pink line) and a variant-adapted vaccine against a heterologous strain (green line). The fourth dose is assumed to be administered 1 year after the third dose, such that estimates for the three-dose group are from 365 days post dose 3 through to 720 days.

Estimates are based on the relationship between NAT and protection obtained from fitting to the Omicron BA.1 variant. **B** Proportion of dose four effectiveness (against mild disease, hospitalisation, and death) that is attributable to receiving any fourth dose (with either the ancestral or variant-adapted vaccine product), relative to the proportion of overall efficacy that is attributable to the variant-adapted product (rather than the ancestral) as the fourth dose, for one year following the fourth dose. This illustrates that administration of any fourth dose generates most of the incremental initial impact, but that the difference between administering the variant-adapted versus the ancestral vaccine becomes more substantial 6–12 months following vaccination.

endpoints during 1 year of follow-up with the ancestral vaccine compared to the variant-adapted vaccine, such that effectiveness against hospitalisation reduces to 59.5% (95% CrI 53.9–62.7%) 365 days following the fourth dose with the ancestral vaccine but remains substantially higher (75.7%, 95% CrI 72.3–80.4%) with the

variant-adapted vaccine. Across a 1-year period, the overall impact is such that just over half of the additional protection is predicted to occur from delivering a 4th booster dose compared to no booster, and the other half from switching to the variant-adapted vaccine rather than continuing with the ancestral vaccine (Fig. 3B).

## Discussion

As the world transitions towards endemic circulation of SARS-CoV-2, there is a need to continue to evaluate COVID-19 vaccine effectiveness against circulating variants of the virus. Our modelling framework presents a method to integrate the insight that has been generated from understanding the utility of NAT as a correlate of clinical protection with larger population-based cohorts of vaccine effectiveness. By fitting a semi-mechanistic model to such data, it is possible to make short-term projections regarding vaccine effectiveness beyond the period of observation that can help to inform ongoing vaccination strategies and in particular the need for regular boosters for the highest risk populations. However, as discussed further below, there are many challenges in doing so given our partial understanding of how immunity (both vaccine- and infection-induced) develops against SARS-CoV-2 and given the ongoing uncertainty in the evolution of the virus.

One of the main challenges with planning future vaccine booster strategies is assessing the potential effectiveness of future variant-specific vaccines. Given the speed with which new variants continue to emerge, coupled with the difficulty in identifying appropriate comparator groups to directly estimate vaccine effectiveness, it is likely that decisions will need to be made based on immunogenicity data. One method for doing this is to apply the fold reductions estimated from immunogenicity data to the relationship inferred against the Wuhan virus in the original paper by Khoury et al.[14]. However, our results show that doing so results in a more pessimistic prediction of vaccine effectiveness than was observed in the English population data. This may be in part due to the uncertainty in the fold reduction given the widespread variation reported across the different laboratory studies. However, it may also indicate that immune responses other than NAT - from both vaccination and through ongoing exposure to the virus in the community - are providing a higher degree of cross-protection against new variants than would be predicted based on NATs alone.

Our results suggest that the value of the ancestral vaccines, whilst providing initial high levels of protection, has gradually been diminished through both waning of protection following the 3rd and subsequent doses and the substantial immune escape presented by the Omicron variant. In combination, these two effects combine to generate a substantial additional estimated benefit of switching 4th and subsequent doses to variant-adapted vaccines—which we estimate to prevent nearly double the number of episodes of severe disease over a 1-year period compared to delivering the 4th dose with the ancestral vaccine. It should be noted that this estimate is sensitive to the underlying shape of our inferred relationship between immune levels and protection, and in particular to the precise point at which there is a rapid drop in protection compared to the "plateau" at high immunity levels. It also depends on the degree to which the new sub-variants that have subsequently emerged exhibit immune escape from the variant-adapted vaccines. However, early follow-up of the impact of bivalent boosters in the US have demonstrated an added benefit over monovalent boosters[21].

One of the major limitations of our work is that it was not possible to distinguish the combined effects of infection- and vaccine-induced immunity. The vaccine effectiveness data to which we fit our model is based on the full population of England with effectiveness estimates obtained by comparing outcomes according to vaccine dose against those with no prior vaccination. Given the widespread circulation of the SARS-CoV-2 virus in the community in England throughout the latter half of 2021 and first half of 2022 (from which these data derive), it is likely that a substantial proportion of both the vaccinated and unvaccinated cohorts will have experienced one or more episodes of infection. This will generate potential biases in vaccine effectiveness as has been recognised across studies attempting to use population level data. This bias will be propagated through to our short-term projections of vaccine effectiveness beyond the observation period. Thus, these patterns of vaccine effectiveness may not hold in other countries with different background levels of infection-induced immunity, and different levels of vaccine mix and uptake over time. Furthermore, there may be additional biases in the data due to other differences in the populations who were not vaccinated, those that received the first two doses but chose not to receive the 3rd dose, and those who received the 3rd dose. These could act to bias the vaccine effectiveness estimates in either direction.

A second limitation in understanding the immune dynamics driving these patterns of vaccine effectiveness was the lack of associated immunological measurements. To overcome this, we inferred immunity levels by treating them as an unobserved process by utilising the parametric forms that have previously been developed to relate NAT to clinical protection. In doing so, we capture the effect of all aspects of the immune response—including both antibody-mediated immunity and potential T-cell responses – in our inferred immunity levels. However, this simple approach, whilst appropriate for short-term parametric projections, may fail to fully capture longer-term immune dynamics. In particular, more complex immune dynamics could generate patterns of decay in immunity levels that diverge from the simple bi-phasic pattern we have assumed here. Furthermore, it does not allow us to gain any further mechanistic insight into the underlying immune dynamics driving the observed vaccine effectiveness against the different clinical endpoints. In particular, we are unable to determine in our analysis whether any single immune marker can be considered a correlate of protection against both mild and severe infection. Further research in this area will require careful analysis of large population cohorts containing both immunological and clinical measurements.

One of the challenges with estimating vaccine effectiveness from population cohort data is the difficulty with distinguishing hospitalisations or deaths arising due to COVID-19 from those arising from other causes but in which patients also received a positive COVID-19 diagnosis. The vaccine effectiveness estimates that we used to fit our model are based on hospital admission data with a recent positive diagnosis and hence do not allow us to disentangle this. However, analyses by UKHSA have shown similar vaccine effectiveness estimates when restricting the data to those that were admitted with a respiratory diagnosis which, whilst imperfect, may be more representative of those admitted due to COVID-19[22]. However, this switch in endpoint also means that it is difficult for us to retrospectively fit all datasets given the changing definition of hospitalisation. This limits our ability to truly assess waning in vaccine effectiveness over time.

Regular booster vaccination is expected to be a key part of the ongoing management of COVID-19 over the coming years, especially among older and more clinically vulnerable populations where protection from any degree of SARS-CoV-2 infection may be crucial. As the SARS-CoV-2 virus continues to evolve, validated models that can estimate the effectiveness of modified vaccine products based on immunogenicity data alone will be increasingly important for assessing the benefit of additional doses with either existing or variant-modified vaccines. Our results demonstrate the challenges with doing so, given the more complex immune landscape that has arisen with both past vaccination and ongoing infection with multiple Omicron sub-variants across the world.

## Methods

### Data

We used empirical estimates of vaccine effectiveness against what we term mild disease (positive *polymerase chain reaction (PCR)* tests including symptomatic cases and asymptomatic infections detected through screening in schools and workplaces), hospitalisations (defined as admission recorded in the Emergency Care Dataset within 14 days of a positive test) and death (within 28 days of a positive test)

with the Delta and Omicron BA.1/BA.2 variants from England. Data were available for three vaccines administered in England—the Oxford/AstraZeneca AZD1222 vaccine, the Pfizer-BioNTech BNT162b2 vaccine, and the Moderna mRNA-1273 vaccine – in various combinations. For our primary analysis we use data from all age groups from the two publications by Andrews et al.[23,24] alongside the estimates in the >65 age group from Stowe et al.[25] in which only age-stratified estimates were provided. We additionally provide results using the <=65 age group in supplementary material. The original publications did not stratify these estimates by sex or gender. For these population-level estimates, data on prior infection was not available and hence the vaccine effectiveness estimates are representative of patterns of protection against a background of partial (and unknown) infection-induced immunity. These vaccine effectiveness estimates may therefore be biased if there are differences between the unvaccinated and vaccinated groups in terms of exposure to the virus.

[12]To test our model projections, we compare our model projected vaccine effectiveness against hospitalisation to more recent estimates of vaccine effectiveness in England over a 15-month period of follow-up[20]. These data are presented only as aggregate estimates across vaccines. Furthermore, the definition used for the endpoint differs as recognition of the issues with incidental infections has been incorporated; thus the endpoint for hospitalisation in these more recent estimates is defined as requiring at least 2 days stay with a respiratory code in the primary diagnosis field. This could be expected to result in slightly higher vaccine effectiveness compared to the definition in the estimates used for our model fitting[26]. These estimates are for the 65+ age-group only. Estimates are available for vaccine effectiveness following dose 2 and dose 3.

## Immunological model

We followed the approach of Khoury et al.[14] by considering the relationship between (here unobserved) immunity level (IL) over time, and protection against mild disease, hospitalisation and death[14]. We first express an individual's IL over time, $n(t)$, as a biphasic exponential decay function where $n_{ij}$ is the initial IL of vaccine $i$ drawn from a $\log_{10}$-normal distribution at dose $j$. Based on B-cell dynamics, we assume an initial period of fast decay with half-life $h_s$ (decay rate $\pi_1 = -\ln(2)/h_s$) representing the combined biochemical decay of antibodies and the ongoing production of antibodies by circulating (mostly short-lived) plasma cells, followed by a second period of slow decay with half-life $h_l$ (decay rate $\pi_2 = -\ln(2)/h_l$), representing ongoing antibody production by long-lived plasma cells. This is represented by

$$n(t) = n_{ij} \frac{\exp(\pi_1 t + \pi_2 t_s) + \exp(\pi_2 t + \pi_1 t_s)}{\exp(\pi_1 t_s) + \exp(\pi_2 t_s)}, \quad (1)$$

where $t$ represents the time since the last dose, and $t_s$ is the period of switching between the fast and slow declines. This results in a smoothed biphasic exponential waning of immunity levels. Whilst the original model is based on a simplification of B-cell dynamics, we note that bi-phasic patterns of immune decay also provide a good approximation to the more complex models of T-cell dynamics against other viruses in relation to longer-term protection following the initial acute infection[27,28].

Following Khoury et al.[14] we assume a logistic relationship between IL and vaccine effectiveness to capture time-varying vaccine protection against infection or mild disease ($m = 1$), hospitalisation ($m = 2$) and death ($m = 3$) given by the function

$$\epsilon_m(n) = \frac{1}{1 + e^{-k\left[\log_{10}(n) - \log_{10}(n_{50_m})\right]}}, \quad (2)$$

where $\epsilon_m$ is vaccine effectiveness, $k$ is the fitted shape parameter and $n_{50_m}$ is the IL relative to convalescents required to provide 50%

protection (from mild disease, hospitalisation, or death)[14]. Under this approach, we estimate different $n_{50_m}$ values for the different endpoints.

Whilst building on the modelling framework proposed by Khoury et al.[14], our approach differs in two ways. First, rather than explicitly using NAT as a correlate, we backwards infer the IL over time from the estimates of vaccine effectiveness. This allows us to capture broader immunity mechanisms if they are not mediated via NAT directly. Second, we simplified the mathematical function for the decay in IL over time, reducing the number of parameters and using a flexible functional form that is able to fit both a single and bi-phasic data. In doing so we are able to infer the pattern of decay that best explains the vaccine effectiveness estimates over time.

An alternative model is one in which protection against severe disease (hospitalisation and death) conditional upon infection or mild disease is assumed to be constant over time[29]. We consider this as an alternative model structure in the Supplementary Appendix.

We considered two different approaches to capture the effect of third and fourth doses. In our main analysis, we consider a vaccine-dependent restoration of IL to a fixed dose-dependent level after each dose. Under this model, the restoration of protection is independent of the past decay in IL. Furthermore, the IL achieved at the 3rd and subsequent doses is independent of the vaccine regime used for the two initial primary course doses, consistent with results from the COV-BOOST trial[30]. As a further exploration of the impact of model structure on our results, we assumed an alternative model in which there is a vaccine- and dose-dependent boost to IL, which therefore restores IL following 3rd and subsequent doses to a level that depends on both the magnitude of the boost and the level of IL achieved post dose 2. Under this model, the IL achieved at 3rd and subsequent doses is therefore dependent on the vaccine regime used for the primary course but by being related to the level achieved post dose 2, does not change according to the time that has elapsed since dose 2. The results from this model are given in the Supplementary Appendix.

## Model fitting

Khoury et al.[14] obtained parameters for this model of vaccine-induced protection by fitting the relationship between NAT data following dose 2 (which are based on the mean 28-day values reported in clinical trials relative to the mean titre for a convalescent individual) to clinical efficacy data from Phase III trials[14]. In order to incorporate the information from this independent set of clinical data, we used their resulting estimates determining the relationship between NAT and protection as informative priors for our Bayesian model fitting. We used data from Wheatley et al.[31] and Pelleau et al.[32] as mildly informative priors for the durability of the immune response over time. In addition, we used immunogenicity data on the NAT observed following the second and third doses from clinical trials to generate informative prior distributions for the three vaccines (Table 1). For the remaining parameters we used wide (uninformative) priors. To fit the model jointly to estimates of vaccine effectiveness against the different variants (Delta and Omicron BA.1/2), we introduce a variant fold reduction (VFR) factor (to represent the degree of immune escape of Omicron) which scales the IL against Omicron compared with the baseline IL against Delta. This parameter is estimated in model fitting.

Our priors and their sources are summarised in Table 1. IL and VFR were transformed to the $\log_{10}$ scale, and all other parameters were fitted on a linear scale. We used Normal distribution priors on these scales with mean and 95% confidence intervals shown in Table 1. The model outputs at the mid-point of the time interval over which the vaccine effectiveness estimates were reported was used in the likelihood function. Model fitting was undertaken using parallel tempered MCMC methods using the DrJacoby R package with 400,000 samples[33].

## Projecting variant-adapted vaccine effectiveness

As new COVID-19 vaccines will be evaluated based on immunogenicity data, we use NAT to provide a preliminary estimate of the effectiveness of variant-adapted vaccines that are now in widespread use. To estimate the benefits for vaccine effectiveness, we used estimates of the relative neutralisation titres of variant-adapted vaccines compared to ancestral vaccines reported by Khoury et al.[16] to explore the potential benefit of variant-adapted vaccines obtained from studies across all age-groups. They estimated a 1.61-fold (95% CI 1.5–1.8) increase in NAT against across all strains compared to the ancestral vaccines, with a higher increase against (1.85, 95% CI 1.6–2.1) against homologous strains and lower increase (1.47, 95% CI 1.3–1.7) against heterologous strains. These data are not age-stratified[16]. This scaling of NAT was sampled from the reported estimates assuming a Normal distribution and then applied (assuming it is representative of the scaling of the broader IL) to the fitted model to generate estimates of vaccine effectiveness against infection, hospitalisation, and death using our model that relates IL to protection from the BA.1/2 variant. We note that there is considerable uncertainty in this scaling, given the variability in immunogenicity responses generated across different laboratories.

**Inclusion and ethics statement.** This work was undertaken by an international team utilising previously published data from the UK. Ethics permission was not sought as all data are in the public domain. Previous versions of this work were regularly shared with UK and international scientific committees to support local COVID-19 response policies.

## Reporting summary

Further information on research design is available in the Nature Portfolio Reporting Summary linked to this article.

## Data availability

The data used in the fitting were extracted from the cited papers into Microsoft® Excel® for Microsoft 365 MSO (Version 2304 Build 16.0.16327.20200) 64-bit. These are provided for ease of access at: https://github.com/mrc-ide/covid_efficacy[34].

## Code availability

All R code (runs performed in version 4.3.0) for this work is available at: https://github.com/mrc-ide/covid_efficacy. All figures and tables can be reproduced using this code.

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

## Acknowledgements

This work was supported by a grant from WHO. A.B.H. acknowledges support from an Australian National Health and Medical Research Council Investigator Grant and Imperial College Research Fellowship. P.W. is supported by an Imperial College Research Fellowship. O.J.W. is supported by a Schmidt Science Fellowship in partnership with the Rhodes Trust. G.C. and A.C.G. acknowledge support from The Wellcome Trust. A.B.H., P.D., P.W., G.C., G.B., S.L.W., N.M.F. and A.C.G. acknowledge funding from the MRC Centre for Global Infectious Disease Analysis (reference MR/R015600/1), funded by the UK Medical Research Council (MRC) and part of the EDCTP2 programme supported by the European Union. This work was additionally supported by the NIHR Health Protection Unit for Modelling and Health Economics (N.M.F.: [NIHR200908]); and philanthropic funding from Community Jameel (PD, NMF).

## Author contributions

Conceptualisation and study design: A.C.G., A.B.H., N.M.F., O.J.W., G.B., P.W., S.L.W.; Vaccine efficacy model fitting: N.M.F., A.C.G., P.D., A.B.H., D.K.; Analysis: A.C.G., A.B.H., J.T., D.O.M., N.M.F., P.D., S.L.W., G.C.; Visualisation: A.B.H., A.C.G.; Writing—original draft: A.C.G., A.B.H., N.M.F., E.M.R.; Writing—review and editing: all authors.

## Competing interests

A.C.G. has participated as a non-renumerated member of a scientific advisory board for Moderna, has received consultancy funding from GSK and Sanofi for activities related to COVID-19 vaccination and is a member of the CEPI scientific advisory board and Gavi Vaccine Investment Strategy steering committee. She has received grant funding from Gavi for COVID-19-related work. A.B.H., P.W. and A.C.G. have previously received consultancy payments from WHO for COVID-19-related work. ABH provides COVID-19 modelling advice to the New South Wales Ministry of Health, Australia. A.B.H. was previously engaged by Pfizer Inc to advise on modelling RSV vaccination strategies for which she received no financial compensation. E.M.R. is a non-remunerated member of the UK Vaccines Network, the UKRI COVID-19 taskforce and the British Society for Immunology Covid-19 taskforce. The remaining authors declare no competing interests.
