## [Peer Review File · Nature Communications]

Estimating long-term vaccine effectiveness against SARS-CoV-2 variants: a model-based approachREVIEWER COMMENTS

Reviewer #1 (Remarks to the Author):

In this manuscript, the authors used a previously published model by Khoury et al. (2021) that links neutralizing antibody titers (NAT) to vaccine effectiveness. They applied the model to newly available reports of mild illness, hospitalization, and death due to delta and omicron variants in England and estimated vaccine effectiveness based on model fitting. Here, they assumed that immunity level is proportional to the antibody titer and described it as a biphasic exponential decay function. Moreover, they used a logistic function to describe relationship between immunity level and vaccine effectiveness. Although the results and methods are clear, their mechanistic rationale is weak and simply not sufficient to provide meaningful information to the reader. My major comments are as follows:

1. The authors quantify the immune response over time by analyzing the changes in antibody levels induced by B-cells. They assumed that the level of immune response (i.e., immunity level in this study) is directly proportional to the amount of antibody produced. However, it is not clear if Khoury's model is applicable to the current situation. Khoury et al suggested that the observed correlation between NAT and vaccine effectiveness was partially based on unproven links with cellular immunity. In addition, various factors such as infection history of SARS-CoV-2 variants and individual vaccination history should be linked with the vaccine effectiveness. This information cannot be inferred from immunogenicity data alone.
2. The author fitted their model to the datasets on effectiveness of "hospitalization" and "death" for both delta and omicron variants which is extremely limited. And they showed that the corresponding estimated vaccine effectiveness are nearly identical. The reliability of the obtained estimation results is questionable.
3. The fitting results presented in Fig. 2, S1 are not perfect and therefore their conclusions are curious. Observed data suggests that the patterns of reduction for vaccine efficacy by the 2nd and 3rd vaccines do not seem to be significantly different (especially, for mild disease case). However, estimated vaccine efficacy predicts significantly different patterns of reduction. In addition, estimated vaccine efficacy suggests that more booster vaccination is rarely required for the Delta variant. Considering that the age of patients in used data is over 65, this estimation result is not realistic.
4. The authors used data from the 65+ age group and predicted their vaccine effectiveness based on model fitting. In fact, there are various studies showing that the vaccine-elicited neutralization against SARS-CoV-2 variants differs by age. Therefore, their conclusion based on several fixed parameter is not generalized.
5. In the model fitting, the authors used MCMC methods and specifically assumed a normal distribution priors. I understand that it is hard to justify the choice. However, for the robustness of analysis results, it is necessary to check whether the estimated posteriors are influenced by the choice of priors. I would suggest the use of uninformative priors.
6. I wonder the assumption of the biphasic decay immunity level without any observation data. In particular, they used data includes both patients with/without infection history, which is expected that significantly affect the immunity level (that is, they might show different decay profiles).
7. In Fig. 2, S1, it is assumed that all patients administered 3 vaccines at the same time. There should be individual variation on vaccination timing and it should affect the estimation of vaccine effectiveness.
8. The authors compared titers of neutralizing antibodies against the omicron BA.1 variant of mRNA.1273.213 and mRNA.1273. Using 1.61 as the scaling value of NAT on average, the bivalent vaccine effectiveness was predicted. However, I expect that the result is very sensitive to the scaling value, so their conclusion based on the fixed parameter is not generalized.

9. In Fig. 3, they showed the model predicted vaccine effectiveness, but I can not understand how they decided the initial value of the effectiveness. And Fig. 3B is difficult to understand.

10. In Fig.2, what is the meaning of "time (days)"? Is it time after the first dose? And they probably use "booster" in different meaning in Fig.2 and Fig.3. This is confusing.

Reviewer #2 (Remarks to the Author):

Summary

Hogan et al. present a very interesting study which uses a model-based approach to estimate long-term vaccine effectiveness against SARS-CoV-2 variants using immunogenicity data. The work is clearly motivated, the approach presented is valid and well documented, the authors make good use of available data to inform their model parameters, and the Discussion clearly highlights some of the key limitations of the work. I think this approach could be very useful to inform the initial approval stages of new variant vaccines to facilitate early estimation of duration of protection before this can be evaluated from e.g., test-negative studies embedded in the community.

In my view, the main limitation of this work is in the conclusions drawn about the likely duration of the resulting protection, based on the available data. The approach is framed as a necessary alternative to using clinical data to approve vaccines due to the widespread levels of infection-induced immunity. However, the population data used by the authors to fit their model was also unable to distinguish between the combined effects of infection- and vaccine-induced immunity, as described in the Discussion. This has potentially profound implications for subsequent estimates of vaccine effectiveness and duration of protection, as previous infection rates are likely to be differential across unvaccinated and vaccinated groups, hence possibly underestimating the effects of vaccination. This is certainly a limitation of many datasets at this stage of the pandemic, rather than specific to this study, however I would recommend the authors frame the paper accordingly, mentioning this in both the abstract and the data description in the methods, as I have highlighted below. I also recommend further documenting the implications of this in the Discussion for the results presented, to make it clearer to readers the limited capacity for robust conclusions on long-term effectiveness of these vaccines at this stage, using only this available data.

In what follows I have highlighted some minor points where I feel edits may be needed, or further clarity provided.

Comments

Abstract

- The opening sentences imply that this study circumvents the issue with of widespread infection-induced immunity. Although this is possible in principle with the method presented, I think a clause should be included in the abstract results to highlight that this was not the case with the data used for the model.

- In reporting the results, I think it would aid interpretation to highlight the number of doses being compared in each instance rather than using the 'boosting' and 'second booster' terminology which we are moving away from. The comparator groups are also not clear from the abstract wording - boosting compared to what? 3 vs 2 doses, 4 vs 3? This should be made clear in the abstract in order to interpret the results.

- The authors report VE against 'mild disease', should this be 'infection and mild disease'?

Introduction

- Paragraph 2: I would clarify that decisions are based on immunogenicity and safety data.

Methods

- Data: I would make it clear at this stage that this is full population data in England where previous infection was not accounted for versus the fourth dose immunogenicity data where it is.
- Data: Were estimates from other metrics of severity available in this data? This could be a useful addition to the supplementary materials if so due to incidental findings, which were a particular concern for Omicron where infection rates were high. I understand this issue is presented in the Discussion, so this is not a concern if not, although it may be helpful to point out that comparisons between Delta and Omicron may be affected by this, as well as changing surveillance over time.
- Data: It doesn't appear as though age-stratified estimates were available in the Chalkias et al. paper? If not, is using both VE estimates in >65 years with immunogenicity data of 4th doses from all adults problematic? Perhaps any implications of this should be mentioned, as we would expect VE estimates in older age groups to be lower than adults of all ages.

Results

- Results are comparing the marginal benefit of an additional dose rather than absolute VE (in the sense that it is compared to unvaccinated). This is the relevant comparator at this stage however the interpretation is not straightforward for policy when 'absolute' values are not presented, as the benefit may be substantial in relative terms but may not necessarily translate to substantial public benefit. I think it is relevant to briefly highlight this at some point in the manuscript if recommending additional doses should be with bivalent vaccines (see e.g., Lewis et al., McMenamin et al.)
- Tables: Not always clear what the comparator group is - include the specific comparisons for VE estimates in each case in the captions of Table 2, Table 3 and Table S1.
- Table 2/Table 3: The vaccine effectiveness estimates against mild disease, hospitalization and death for the Moderna mRNA.1273 vaccine presented over time are the same post 3rd dose (Table 2) and post 4th dose (Table 3). Is this a typo in generating the tables or have I misunderstood? This would also need to be changed in the results text.
- Figure 3B: Again, the terminology is not consistent. The comparison of 'no boosting' appears to be no further 'boosting' after the third dose however elsewhere in the manuscript the third dose is described as a 'booster' and the fourth a 'second booster'. This needs to be standardized.

Lewis NM, Chung JR, Uyeki TM, Grohskopf L, Ferdinands JM, Patel MM. Interpretation of Relative Efficacy and Effectiveness for Influenza Vaccines. *Clin Infect Dis*. 2022 Aug 24;75(1):170-175. doi: 10.1093/cid/ciab1016. PMID: 34875035.

McMenamin ME, Bond HS, Sullivan SG, Cowling BJ. Estimation of Relative Vaccine Effectiveness in Influenza: A Systematic Review of Methodology. *Epidemiology*. 2022 May 1;33(3):334-345. doi: 10.1097/EDE.0000000000001473. PMID: 35213508; PMCID: PMC8983951.

Discussion

- Paragraph 4: Rather than these estimates may not hold in other countries, I would suggest that bias may have been introduced not only due the number of previous infections being unknown but

because of the possibility that this differs by vaccine status, by vaccine type and over time.

- Paragraph 6: I would also highlight that England has very good vaccine coverage, particularly in the >65 age group, and that those vaccinated with more doses may differ systematically to those vaccinated with fewer doses, potentially introducing bias in an unknown direction (healthy vaccinee effect where those receiving more doses may be more health conscious? Or those receiving more doses may be immunosuppressed and more worried about COVID?). If this is not accounted for in the original study this would affect estimates using the severe disease definition presented in this work.

Typos

- Page 6, final paragraph: space needed between 'BNT162b2' and 'and'

Reviewer #3 (Remarks to the Author):

This paper by Hogan et al addresses a critical question which is estimating SARS-CoV-2 vaccine efficacy in the complex mixed immune environment with multiple co-circulating variants and different levels of prior infection and vaccination (1st and 2nd boosters). The paper utilizes a sensible mathematical model which bridges a non-measurable immune level with vaccine efficacy against infection, hospitalization, and death. The modeling is adequate for this task. The analyses are reasonable and sound, the figures are clear and the paper is quite well written. The paper will be of interest to epidemiologists, immunologists and public health officials. There are a few issues to address where the claims of the paper exceed what is possible with modeling:

1) Most critically, there are multiple reasons why any projection of vaccine efficacy against hospitalization &/or death are deeply uncertain. The authors do a nice job of outlining these reasons but do not acknowledge the resulting massive degree of uncertainty of their projections. To summarize, neutralizing antibody levels are not likely to be a great surrogate for disease severity for which there are no precisely identified immunologic surrogates. One reasonable possibility are tissue-resident T cells which follow different dynamics than antibodies and circulating T cells. The model assumption that B and T cells follow similar dynamics is also risky and likely pathogen dependent. Second, the models projecting well beyond the observed data, deep into a period of immune memory uncertainty. Any projection beyond observed data requires validation before being used for predictive purposes. Third, as the authors describe well, COVID-19 related hospitalization is now a misclassified outcome and therefore the data for model fitting may be incorrect. Fourth, the authors' "alternative model" of fixed hospitalization rate given infection fits the data the best but they do not use it as the "main" model based on a somewhat vague reference to a published paper (ref 27). In general, superior fitting models should be used for projection but conflicting data in the literature only highlights the fact that our pre-existing knowledge on correlates of disease severity is inadequate. Fifth, it is impossible to predict the immune evasion properties of new CoV-2 variants as they pertain to severe outcomes. When all these factors are considered together, it seems like a more measured and safe approach is to acknowledge that it is impossible to predict vaccine efficacy against hospitalization and death beyond a certain very short timeframe. I would be more comfortable with these outcomes presented in the supplement with acknowledgement that they are highly exploratory. The projections of protection against infection seem somewhat safer. As an aside, I beg the authors to produce a follow up paper in a year or so when the competing models can be tested and compared against more extended longitudinal data.

2) For protection against infection, why is the analysis restricted to age > 65?

3) In Fig 1, S2 and S3, please be more explicit about what on the graphs is real data versus model output. Are the 95% credible intervals related to model output and why then are they only applied to one model and not the other. It was difficult for me to compare models for fit to data based on

these graphs.

4) The methods for obtaining different dose-response curves in Figure 1 in Khoury et al versus the more pessimistic curves in the present paper are a bit confusing. Please consider an added methods figure to demonstrate how the 2 papers arrive at different curves. The second paragraph of the discussion which tries to explain this divergence is the one paragraph in the paper that confused me a bit and could use a re-write.

5) A small point.... I am not sure comparative model fitting is really a form of sensitivity analysis.

6) Another small point is that the model is not truly mechanistic given that it does not link to cellular dynamics in a precise way. I would say semi-mechanistic at best.

REVIEWER COMMENTS

Reviewer #1 (Remarks to the Author):

In this manuscript, the authors used a previously published model by Khoury et al. (2021) that links neutralizing antibody titers (NAT) to vaccine effectiveness. They applied the model to newly available reports of mild illness, hospitalization, and death due to delta and omicron variants in England and estimated vaccine effectiveness based on model fitting. Here, they assumed that immunity level is proportional to the antibody titer and described it as a biphasic exponential decay function. Moreover, they used a logistic function to describe relationship between immunity level and vaccine effectiveness. Although the results and methods are clear, their mechanistic rationale is weak and simply not sufficient to provide meaningful information to the reader. My major comments are as follows:

1. The authors quantify the immune response over time by analyzing the changes in antibody levels induced by B-cells. They assumed that the level of immune response (i.e., immunity level in this study) is directly proportional to the amount of antibody produced. However, it is not clear if Khoury's model is applicable to the current situation. Khoury et al suggested that the observed correlation between NAT and vaccine effectiveness was partially based on unproven links with cellular immunity. In addition, various factors such as infection history of SARS-CoV-2 variants and individual vaccination history should be linked with the vaccine effectiveness. This information cannot be inferred from immunogenicity data alone.

Whilst we appreciate the points made by the reviewer regarding the wider immune responses that are likely to be underlying vaccine effectiveness, in this work we deliberately sought not to rely directly on neutralizing antibody data, but rather to fit a model to the clinical data on vaccine effectiveness. To do so, we infer “immunity levels” which translate to protection through a dose response curve. This mechanism has its origins in pharmacokinetic/pharmacodynamic modelling for therapeutics, where drug levels are related to efficacy through a similar dose-response curve. The model is therefore agnostic as to the underlying mechanisms of immune protection. We therefore disagree that the “mechanistic basis for doing so” is weak.

The only resulting choice to be made is in the functional shape of the change in immunity levels over time following vaccination. Here we have chosen to use a bi-phasic pattern that replicates the patterns that would be expected from a broader model of B-cell dynamics but that is also similar (other than the initial priming phase) to patterns generated by models of T-cell dynamics (see references 22-23 in the manuscript). However, we agree that there are limitations and we have expanded our discussion to highlight these as well as reviewed the text throughout to ensure that our results are presented with the appropriate caveats.

We agree that there are further complexities and heterogeneities that determine vaccine effectiveness – including SARS-CoV-2 variants, vaccination history and infection history. The first two in this list are accounted for in our analysis; the published estimates of vaccine effectiveness that we fit our model to are stratified by SARS-CoV-2 variants (Delta and Omicron respectively), and by COVID-19 vaccination history. The latter – infection history – is difficult to assess and this potential bias is discussed extensively in the original publication of the vaccine effectiveness estimates (and applies to all estimates of vaccine effectiveness against SARS-CoV-2 obtained from real-world data). This limitation was included in paragraph 3 of the discussion, but we have expanded this as well as considered the text throughout the manuscript to ensure that this is clear.

The reviewer states that the Khoury et al. correlation is based on “unproven links” with cellular immunity. This is not the case; what that paper demonstrated is that, using clinical trial data, there is a clear correlation between neutralizing antibody titres and clinical protection afforded by vaccination. What we do here is to demonstrate that a similar functional form can be used to fit the observed vaccine effectiveness over time across a full population cohort. This does not mean that cellular immunity would necessarily take the same functional form, nor that neutralizing antibody titres are linked to cellular immunity. However, it does demonstrate that this functional form of decay in immunity levels is able to reproduce observed patterns of vaccine effectiveness. Thus we partially move forward the debate as to whether neutralizing antibody titre acts as a surrogate for levels of protection, but do not exclude additional cellular immunity mechanisms playing a role in the observed levels of protection.

2. The author fitted their model to the datasets on effectiveness of “hospitalization” and “death” for both delta and omicron variants which is extremely limited. And they showed that the corresponding estimated vaccine effectiveness are nearly identical. The reliability of the obtained estimation results is questionable.

We disagree that the data are limited; the vaccine effectiveness estimates arise from linked data analysis from the English population – some 50 million individuals – and so it would be difficult to think of a larger study dataset. Any residual uncertainty is captured in the uncertainty intervals around the data that are incorporated in our data likelihood used to fit the model.

The availability of data also depends on the vaccine dose-mix delivered in England, as well as the timing of booster doses in relation to the emergence of the variants. Considering the full dataset (originally shown in Figures 2 and S1) there are relatively precise estimates of vaccine effectiveness following dose 2 against the Delta variant for both hospitalisation and deaths for AZ-AZ, AZ-PF, AZ-MD, PF-PF and PF-MD vaccine combinations. As these are post dose 2, these are the VE estimates for AZ and PF which were the two vaccines most widely used in this population. The lack of estimates following the booster reflects the timing of boosters compared to the replacement of the Delta variant by the Omicron variant. To make this clearer, we have modified Figure 2 to include all schedules.

Similarly, against the Omicron variant, there are precise estimates of VE against hospitalisation post dose 3 for those that were boosted with either PF or MD. AZ was not widely used for boosting, and again the timing of the booster programme coincided with the emergence of Omicron. The lack of estimates against the death endpoint reflects the lower severity of the Omicron variant such that there have been few events (despite this remaining a population-wide analysis).

There is no a priori reason for us to consider that the protection against hospitalisations and deaths should not be the same. There is certainly sufficient power in these data to identify differences against the Delta variant – these were not observed in the original study and our model fitting reflects this.

We therefore disagree that the model estimates are questionable – they reflect the patterns observed in the data. Whilst all model fits are imperfect, given the simplicity of the model, we feel this provides a parsimonious fit to the data. This is particularly the case given some of the known biases in the data that we acknowledge as a limitation in the discussion - see also responses to Reviewer 2 and 3.

3. The fitting results presented in Fig. 2, S1 are not perfect and therefore their conclusions are curious. Observed data suggests that the patterns of reduction for vaccine efficacy by the 2nd and 3rd vaccines do not seem to be significantly different (especially, for mild disease case). However, estimated vaccine efficacy predicts significantly different patterns of reduction. In addition, estimated vaccine efficacy suggests that more booster vaccination is rarely required for the Delta variant. Considering that the age of patients in used data is over 65, this estimation result is not realistic.

Thank you for pointing this out. Having reviewed the figures we realised that there was a plotting error that was the cause of the different patterns of decay after the 2nd and 3rd doses. This has now been rectified.

The model results do suggest that further booster vaccinations would not have been required if the Delta variant had remained in circulation. Of course, this is now impossible to verify.

See below our response regarding the age of individuals in the study.

4. The authors used data from the 65+ age group and predicted their vaccine effectiveness based on model fitting. In fact, there are various studies showing that the vaccine-elicited neutralization against SARS-CoV-2 variants differs by age. Therefore, their conclusion based on several fixed parameter is not generalized.

We chose to present the data for individuals over 65 as the patterns observed in the original study in those under 65 are very similar but the numbers of events are larger in the over 65 group due to their higher risk. We note that most of the published data are from all age-groups; it is only the later data on the Omicron variant published in Stowe et al. 2022 for which estimates were only provided stratified by the 18-64 and 65+ age-groups. We have now made this clearer in the methods section. We have also included the comparable fits and estimates using the Stowe et al. VE for 18-64 years in the supplementary information.

5. In the model fitting, the authors used MCMC methods and specifically assumed a normal distribution priors. I understand that it is hard to justify the choice. However, for the robustness of analysis results, it is necessary to check whether the estimated posteriors are influenced by the choice of priors. I would suggest the use of uninformative priors.

We adopted a Bayesian framework and incorporated mildly informative priors (Normal on a log scale) specifically so that external relevant data could be included in our analysis. The priors for doses 2 and 3 of each vaccine are based on immunogenicity data. The priors for the immunity function parameters are based on those obtained by Khoury et al. (who fitted a similar model to independent data from the original clinical trials) and from other data sources on the decay profile of neutralizing antibody titres. These sources are referenced in Table 1. The methods section has been updated and expanded to make this clearer. We have also slightly modified the priors and the presentation in Table 1 to make the link to the prior data clearer. This has had a small effect on the outputs which have all been updated.

6. I wonder the assumption of the biphasic decay immunity level without any observation data. In particular, they used data includes both patients with/without infection history, which is expected that significantly affect the immunity level (that is, they might show different decay profiles).

As noted earlier, we chose to model the immunity levels as a biphasic decay pattern based on the wider literature on both B-cell and T-cell dynamics. Whilst it is possible that prior infection could affect the maturation stage of SARS-Cov-2 specific cells and subsequently modify this function, in the absence of strong evidence we feel that this is a parsimonious assumption to make. However, in modifying our priors we also took the opportunity to use relatively uninformative prior for the second phase. By doing so, the fitting can “choose” a single decay rather than a biphasic decay if this fits the data better. The fact that the resulting profile remains bi-phasic suggests that this is an appropriate form, at least given the priors for other model parameters that have been informed by other data sources.

It is not possible using these data to infer differences between those who have previous infection-induced immunity versus those who have only vaccine-induced immunity. As the vaccine effectiveness estimates compared the unvaccinated to those vaccinated, in the early phase of the data we would expect these groups to have similar levels of prior exposure to the virus (and hence similar levels of infection-induced immunity). However, as noted in the original publication of these vaccine effectiveness estimates, there will likely be a bias resulting from higher infection levels in the unvaccinated given that the vaccines provide partial protection against infection. This would result in lower estimated vaccine effectiveness than the true underlying vaccine efficacy. This is a limitation in all observational vaccine effectiveness studies; as noted earlier, we have included this in the discussion.

7. In Fig. 2, S1, it is assumed that all patients administered 3 vaccines at the same time. There should be individual variation on vaccination timing and it should affect the estimation of vaccine effectiveness.

We do not have individual level data on the precise timing of vaccination, but this was available in the original data analysis. In the original publication of vaccine effectiveness, the data are presented grouped into days since dose 1/2/3. We fit the model using the mid-point of this interval. The precise estimates will of course be sensitive to this choice but without further data on the distribution of observations across the interval, this seems an appropriate choice to make. This is now noted in the methods section of the manuscript.

8. The authors compared titers of neutralizing antibodies against the omicron BA.1 variant of mRNA.1273.213 and mRNA.1273. Using 1.61 as the scaling value of NAT on average, the bivalent vaccine effectiveness was predicted. However, I expect that the result is very sensitive to the scaling value, so their conclusion based on the fixed parameter is not generalized.

We agree that the scaling factor is highly dependent on the assays used in this particular study. We intended this to be an illustration of the approach, rather than as a clear prediction. Furthermore, in practice it will be necessary to use multiple assay measurements to determine the sensitivity to this. We have now noted this in the methods and retained this in the discussion (paragraph 3).

9. In Fig. 3, they showed the model predicted vaccine effectiveness, but I can not understand how they decided the initial value of the effectiveness. And Fig. 3B is difficult to understand.

We apologise if this was unclear. The time-scale on the x-axis in Figure 3 is now days since the administration of the 4th dose, which we assume here to occur 1 year following the 3rd

dose. The initial value for the scenario with no further boosting is therefore 365 days following dose 3, whilst for the monovalent and bivalent fourth dose vaccines it is the model-predicted vaccine effectiveness immediately following the administration of the fourth dose. This has now been added to the legend.

This figure was originally developed to inform WHO considerations of the value of 3rd doses in LMIC. However, given that many countries subsequently administered bivalent vaccines as fourth doses, we have switched this to represent a fourth dose compared to a third dose.

Figure 3b shows the difference between administering a 4th dose with the monovalent vaccine compared to not administering a 4th dose (i.e. the difference between the orange and green lines in Figure 3a) in orange, and the difference between using the bivalent vaccine versus the monovalent vaccine (i.e. difference between the purple and orange lines in Figure 3a) in purple, as a percentage of the total increase in VE that can be achieved with administering a 4th dose with the bivalent vaccine (i.e. difference between the purple and green lines in Figure 3a). This panel is included to illustrate that, whilst there may not be much apparent additional impact of the bivalent vaccine compared to the original vaccine shortly after administration of the 4th dose, because of the decay dynamics, the impact of a full year is more substantial. The figure was specifically developed to help to communicate this concept to a non-modelling audience. We have modified the text in the legend and the results section to make this clearer.

10. In Fig.2, what is the meaning of “time (days)”? Is it time after the first dose? And they probably use “booster” in different meaning in Fig.2 and Fig.3. This is confusing.

We apologise for the confusion here. For clarity, throughout the manuscript, we have modified our language to remove “booster” and refer instead to dose numbers as we agree this is clearer.

Reviewer #2 (Remarks to the Author):

Summary

Hogan et al. present a very interesting study which uses a model-based approach to estimate long-term vaccine effectiveness against SARS-CoV-2 variants using immunogenicity data. The work is clearly motivated, the approach presented is valid and well documented, the authors make good use of available data to inform their model parameters, and the Discussion clearly highlights some of the key limitations of the work. I think this approach could be very useful to inform the initial approval stages of new variant vaccines to facilitate early estimation of duration of protection before this can be evaluated from e.g., test-negative studies embedded in the community.

In my view, the main limitation of this work is in the conclusions drawn about the likely duration of the resulting protection, based on the available data. The approach is framed as a necessary alternative to using clinical data to approve vaccines due to the widespread levels of infection-induced immunity. However, the population data used by the authors to fit their model was also unable to distinguish between the combined effects of infection- and vaccine-induced immunity, as described in the Discussion. This has potentially profound implications for subsequent estimates of vaccine effectiveness and duration of protection, as previous infection rates are likely to be differential across unvaccinated and vaccinated groups, hence possibly underestimating the effects of vaccination. This is certainly a limitation of many datasets at this stage of the pandemic, rather than specific to this study, however I would recommend the authors frame the paper accordingly, mentioning this in both the abstract and the data description in the methods, as I have highlighted below. I also recommend further documenting the implications of this in the Discussion for the results presented, to make it clearer to readers the limited capacity for robust conclusions on long-term effectiveness of these vaccines at this stage, using only this available data.

In what follows I have highlighted some minor points where I feel edits may be needed, or further clarity provided.

We thank the reviewer for their thoughtful and constructive comments. With these in mind, we have reviewed the way that we present the work and taken on board many of the suggestions as detailed below. Given the uncertainty in the estimates of vaccine effectiveness beyond the follow-up time in the dataset, we have decided to remove these from the abstract so that they can be presented with this uncertainty in mind in a more detailed way in the main text.

However, we also note that data are now available from UKHSA with follow-up to one year following dose 3. Whilst this cannot formally be included in our fitting because of changes in the way in which the analyses were performed, it does provide a degree of confidence that our projections to 1 year are sensible. This is now included in the manuscript.

Comments

Abstract

- The opening sentences imply that this study circumvents the issue with of widespread infection-induced immunity. Although this is possible in principle with the method presented,

I think a clause should be included in the abstract results to highlight that this was not the case with the data used for the model.

We agree and have modified the abstract in relation both to the wordings around “projections” beyond the point at which data were available, and to note the limitation that these estimates may change with widespread infection-induced immunity.

- In reporting the results, I think it would aid interpretation to highlight the number of doses being compared in each instance rather than using the ‘boosting’ and ‘second booster’ terminology which we are moving away from. The comparator groups are also not clear from the abstract wording - boosting compared to what? 3 vs 2 doses, 4 vs 3? This should be made clear in the abstract in order to interpret the results.

We agree and have changed this throughout.

- The authors report VE against ‘mild disease’, should this be ‘infection and mild disease’?

As noted in the methods, the VE estimates include both symptomatic cases and asymptomatic infections detected through screening in schools and workplaces. Whilst these could be referred to as infection and mild disease, we feel this would be misleading as it is likely that the data were dominated by symptomatic testing. We do however now clarify this in the methods section.

Introduction

- Paragraph 2: I would clarify that decisions are based on immunogenicity and safety data.

Thanks for pointing this out – we have included this.

Methods

- Data: I would make it clear at this stage that this is full population data in England where previous infection was not accounted for versus the fourth dose immunogenicity data where it is.

Thanks -we have included this.

- Data: Were estimates from other metrics of severity available in this data? This could be a useful addition to the supplementary materials if so due to incidental findings, which were a particular concern for Omicron where infection rates were high. I understand this issue is presented in the Discussion, so this is not a concern if not, although it may be helpful to point out that comparisons between Delta and Omicron may be affected by this, as well as changing surveillance over time.

The published vaccine effectiveness data include a number of methods to estimate vaccine effectiveness. In terms of incidental infections in particular, the UKHSA modified their definition of hospitalisations in later publications to include criteria related to admission from respiratory diseases (e.g. in Stowe et al.). However, this information is not available for the historical estimates and so we cannot piece together a single dataset across the Delta and Omicron variants in order to modify the definition. We have however added further

discussion of this, as it remains a valid criticism of early vaccine effectiveness estimates that we are unable to account for in our model fitting.

- Data: It doesn't appear as though age-stratified estimates were available in the Chalkias et al. paper? If not, is using both VE estimates in >65 years with immunogenicity data of 4th doses from all adults problematic? Perhaps any implications of this should be mentioned, as we would expect VE estimates in older age groups to be lower than adults of all ages.

The reviewer is correct in noting that the estimates in Chalkias et al. are not age-stratified but the summary characteristics show approximately 60% of participants were 18-64 and 40% were 65 years and above. This has now been noted in the methods.

As noted in our response to reviewer 1, the VE data that we fit to is mostly all age. It is only the later VE estimates in Stowe et al. 2022 that were provided stratified (and not aggregated) and hence we fitted our model to the 65+ data as there are more events in this group. We have made this clearer in the methods section. We have also included the fits using the 18-64 age group from Stowe et al. in the Supplementary information.

Whilst there is little difference in our fits, we feel that the 65+ group are more relevant going forwards as they are the highest risk group and will be prioritised for future booster vaccination campaigns.

Results

- Results are comparing the marginal benefit of an additional dose rather than absolute VE (in the sense that it is compared to unvaccinated). This is the relevant comparator at this stage however the interpretation is not straightforward for policy when 'absolute' values are not presented, as the benefit may be substantial in relative terms but may not necessarily translate to substantial public benefit. I think it is relevant to briefly highlight this at some point in the manuscript if recommending additional doses should be with bivalent vaccines (see e.g., Lewis et al., McMenamin et al.)

The outputs in Tables 2 and 3 are the absolute estimates of vaccine effectiveness compared to no vaccination. However, the reviewer is correct in noting that the marginal benefits of additional doses are now calculated for dose 4 onwards (for doses 2 and 3 the comparator group in the England vaccine effectiveness estimates was no vaccine; this differs elsewhere). In order to compare our predicted estimates against more recent estimates of dose 4 impact, we have additionally added the relative vaccine effectiveness of dose 4 compared to only receiving 3 doses in Table 3.

- Tables: Not always clear what the comparator group is - include the specific comparisons for VE estimates in each case in the captions of Table 2, Table 3 and Table S1.

The comparator group throughout is no vaccination – we have clarified this in the legends to Tables 2, 3, S1 and S3.

- Table 2/Table 3: The vaccine effectiveness estimates against mild disease, hospitalization and death for the Moderna mRNA.1273 vaccine presented over time are the same post 3rd dose (Table 2) and post 4th dose (Table 3). Is this a typo in generating the tables or have I misunderstood? This would also need to be changed in the results text.

We are assuming that the 4th dose acts in the same way as the 3rd dose (since we did not have vaccine effectiveness estimates available at the time of fitting to suggest otherwise). Thus in our predictions we expect the vaccine effectiveness with mRNA.1273 to return to the levels presented in Table 2 after dose 3. We apologise that this was not clear; we have added this to the methods and to the Table 3 legend.

- Figure 3B: Again, the terminology is not consistent. The comparison of 'no boosting' appears to be no further 'boosting' after the third dose however elsewhere in the manuscript the third dose is described as a 'booster' and the fourth a 'second booster'. This needs to be standardized.

We have standardized this throughout the manuscript.

Lewis NM, Chung JR, Uyeki TM, Grohskopf L, Ferdinands JM, Patel MM. Interpretation of Relative Efficacy and Effectiveness for Influenza Vaccines. Clin Infect Dis. 2022 Aug 24;75(1):170-175. doi: 10.1093/cid/ciab1016. PMID: 34875035.

McMenamin ME, Bond HS, Sullivan SG, Cowling BJ. Estimation of Relative Vaccine Effectiveness in Influenza: A Systematic Review of Methodology. Epidemiology. 2022 May 1;33(3):334-345. doi: 10.1097/EDE.0000000000001473. PMID: 35213508; PMCID: PMC8983951.

Discussion

- Paragraph 4: Rather than these estimates may not hold in other countries, I would suggest that bias may have been introduced not only due the number of previous infections being unknown but because of the possibility that this differs by vaccine status, by vaccine type and over time.

Thanks for the helpful suggestion – we have modified paragraph 4 accordingly.

- Paragraph 6: I would also highlight that England has very good vaccine coverage, particularly in the >65 age group, and that those vaccinated with more doses may differ systematically to those vaccinated with fewer doses, potentially introducing bias in an unknown direction (healthy vaccinee effect where those receiving more doses may be more health conscious? Or those receiving more doses may be immunosuppressed and more worried about COVID?). If this is not accounted for in the original study this would affect estimates using the severe disease definition presented in this work.

We agree but do not have data that can be used to test this. We have added this point to the discussion.

Typos

- Page 6, final paragraph: space needed between 'BNT162b2' and 'and'

Corrected

Reviewer #3 (Remarks to the Author):

This paper by Hogan et al addresses a critical question which is estimating SARS-CoV-2 vaccine efficacy in the complex mixed immune environment with multiple co-circulating variants and different levels of prior infection and vaccination (1st and 2nd boosters). The paper utilizes a sensible mathematical model which bridges a non-measurable immune level with vaccine efficacy against infection, hospitalization, and death. The modeling is adequate for this task. The analyses are reasonable and sound, the figures are clear and the paper is quite well written. The paper will be of interest to epidemiologists, immunologists and public health officials. There are a few issues to address where the claims of the paper exceed what is possible with modeling:

1) Most critically, there are multiple reasons why any projection of vaccine efficacy against hospitalization &/or death are deeply uncertain. The authors do a nice job of outlining these reasons but do not acknowledge the resulting massive degree of uncertainty of their projections.

We agree that this was not clearly articulated; we have modified the text to make this clearer throughout including in the abstract and discussion.

To summarize, neutralizing antibody levels are not likely to be a great surrogate for disease severity for which there are no precisely identified immunologic surrogates. One reasonable possibility are tissue-resident T cells which follow different dynamics than antibodies and circulating T cells. The model assumption that B and T cells follow similar dynamics is also risky and likely pathogen dependent.

As noted in our response to reviewer 1, we opted to fit a bi-phasic decay model that captures a pattern similar to that predicted by models of B-cell dynamics (following from the correlation noted by Khoury et al. between neutralizing antibody titres and protection) but also consistent with patterns of circulating T-cells. We agree that this may not represent tissue-resident T cells and that this may impact our assumed functional form of decay. This is now noted in the discussion. However, at the same time, it is necessary to make short-term projections in order to inform programmatic implementation of further booster campaigns. Having some estimates, even with associated uncertainty, is better in our view than having none. These can be discussed in light of the associated uncertainties.

Second, the models projecting well beyond the observed data, deep into a period of immune memory uncertainty. Any projection beyond observed data requires validation before being used for predictive purposes.

As noted above, it is necessary to make some assumptions about longer term vaccine effectiveness beyond the period of observation to inform discussion of future boosting scenarios. We have been cautious in our presentation of these results, and have further modified the text throughout the manuscript to reflect this uncertainty, as well as to be clear as to when the estimates go beyond the data used in the fitting (now shaded in Table 2 and made clear that Table 3 is entirely projections).

However, given that several months have now passed since we first performed this analysis, it is now possible for us to compare our "predictions" to the "observed vaccine effectiveness" that has since been released by UKHSA up to >1 year following the third dose. We have added this comparison to the results. For several reasons (including definitions of hospitalisations as noted below), it is not possible to make a direct comparison. However, we

feel that the close agreement between our predicted vaccine effectiveness and those observed adds strength to the validity of our approach and provides some reassurance regarding the predictive power of the model over this timeframe of 1 year.

Third, as the authors describe well, COVID-19 related hospitalization is now a misclassified outcome and therefore the data for model fitting may be incorrect.

As noted in our response to reviewer 2, the published vaccine effectiveness data include a number of definitions of hospitalisation. In terms of incidental infections in particular, the UKHSA did modify their definition of hospitalisations in later publications to include criteria related to admission from respiratory diseases (e.g. in Stowe et al.). However, this information is not available for the historical estimates and so we cannot piece together a single dataset across the Delta and Omicron variants in order to modify the definition. We have however added further discussion of this, as it remains a valid criticism of early vaccine effectiveness estimates that we are unable to account for in our model fitting.

Fourth, the authors' "alternative model" of fixed hospitalization rate given infection fits the data the best but they do not use it as the "main" model based on a somewhat vague reference to a published paper (ref 27). In general, superior fitting models should be used for projection but conflicting data in the literature only highlights the fact that our pre-existing knowledge on correlates of disease severity is inadequate.

The alternative model was generated as a sensitivity analysis. The reviewer is correct that one would normally select the best fitting model if only the current data were available. However, at the time that the manuscript was submitted for review, there were longer-term follow-up data reported in reference 27 that demonstrate that the alternative model would now be a poorer fit. Given the different way in which these data were reported (aggregate across vaccine combinations) it was not possible for us to use this directly to update our fitting. We have made this clearer in the text.

As noted earlier, the projections from this model out to 1 year following dose 3 remain consistent with the observations that have since been released. We feel this demonstrates the validity of the approach.

Fifth, it is impossible to predict the immune evasion properties of new CoV-2 variants as they pertain to severe outcomes.

We agree but this is not the purpose of this exercise. It is however important to understand retrospectively what the impact was of the variants that have arisen, which is what we consider here.

When all these factors are considered together, it seems like a more measured and safe approach is to acknowledge that it is impossible to predict vaccine efficacy against hospitalization and death beyond a certain very short timeframe. I would be more comfortable with these outcomes presented in the supplement with acknowledgement that they are highly exploratory.

The projections of protection against infection seem somewhat safer. As an aside, I beg the authors to produce a follow up paper in a year or so when the competing models can be tested and compared against more extended longitudinal data.

Whilst we agree with the reviewer that the future is impossible to predict, it is equally impossible to make decisions about future strategies without some model of how these might impact. The purpose of the framework presented here was to test and validate a simplified model of the relationship between immune levels and protection, in order to make

short-term projections of the shape of decay in vaccine effectiveness against different endpoints. These projections have now been shown to be consistent with observed vaccine effectiveness data which we feel provides a degree of validation of the approach.

This remains an area of active ongoing research and we will certainly continue to explore ways to update and test these models going forwards.

2) For protection against infection, why is the analysis restricted to age > 65?

As noted above, most of the data were available across all ages (in the papers by Andrews et al.). However, the later Omicron estimates presented in Stowe et al. are only presented disaggregated by vaccine type and age. We chose to use the 65+ age-group estimates here as there were more events in this age-group. We have also now included the results from alternatively using the 18-64 age-group in the supplementary material. These results are very similar and so do not change our conclusions.

3) In Fig 1, S2 and S3, please be more explicit about what on the graphs is real data versus model output. Are the 95% credible intervals related to model output and why then are they only applied to one model and not the other. It was difficult for me to compare models for fit to data based on these graphs.

We have amended the legend to make this clearer. The lines and 95% credible intervals are the inferred relationship between immunity levels and protection from our model fitting (with the model fits shown in Figures 2 and S1). The dashed line is the central fit for the Khoury et al. model which was fitted to data on NAT and clinical endpoints against the Wuhan virus, and then adjusted using immunogenicity data to predict the relationship against the Delta and Omicron variants.

4) The methods for obtaining different dose-response curves in Figure 1 in Khoury et al versus the more pessimistic curves in the present paper are a bit confusing. Please consider an added methods figure to demonstrate how the 2 papers arrive at different curves. The second paragraph of the discussion which tries to explain this divergence is the one paragraph in the paper that confused me a bit and could use a re-write.

Noted. We have extended the results paragraph and legend to Figure 1 to make this clearer. The Khoury et al. paper uses data on NATs and clinical endpoints from the trial. In our fitting, we use a similar model structure, but fit this only to vaccine effectiveness data at the population level. We therefore infer immunity levels that represent both NATs and any additional protection provided via cellular immunity (conditional on the mathematical assumptions made in the model).

5) A small point.... I am not sure comparative model fitting is really a form of sensitivity analysis.

We refer to this as a sensitivity analysis in that it is evaluating the sensitivity to model structure rather than parameters. We have re-worded this to be clear.

6) Another small point is that the model is not truly mechanistic given that it does not link to

cellular dynamics in a precise way. I would say semi-mechanistic at best.

We have modified this to call it semi-mechanistic.

REVIEWER COMMENTS

Reviewer #1 (Remarks to the Author):

The authors have provided a reasonable response to the comments I made regarding their manuscript, particularly with regards to my previous comments 2, 3, 4, 5, 6, 7, 9 and 10. However, I still have two concerns with regards to my previous comments 1 and 8.

1. The authors did not explicitly indicate in their rebuttal letters which part of the main text (e.g., p.XX, line XX-XX) corresponds to our comments, although I may have missed their response. In their reply to my previous comment 1, the authors explained the differing assumptions between their model and the Khoury model; however, I could not find a thorough discussion of this in the main text. I believe it would be beneficial to add this discussion to prevent confusion or misunderstandings among non-modelers.

2. I am still wondering about the fixed value of "1.61" in my previous comment 8. In order to validate the strong assumption that 1.61 may not be essential to reach the same conclusion, the authors should conduct a sensitivity analysis with different values and report the results in the Supplementary Information at a minimum.

Reviewer #2 (Remarks to the Author):

The authors have provided a thorough revision addressing all comments. I recommend that this work is published in Nature Communications.

Reviewer #3 (Remarks to the Author):

The authors have done a great job overall of responding to prior critiques in the paper and in particular acknowledging why the model is not likely to be predictive. The revised discussion is fantastic. I have a few other minor thoughts.

1) The authors state in their response that the purpose of the paper is not to model the immune evasion properties of new variants. This is a fair point but in this sense the model can never be predictive because subsequent waves have a high likelihood of being due to variants whose immune properties are unknown and whose match against existing vaccines is also unknown. It therefore seems necessary to eliminate the terms "prediction" and "predictive" from the paper in favor of "project" and "forecast". Overall, the authors have done a nice job of softening language and providing caveats to the model's ability to project vaccine efficacy over the long term.

2) The assumption that NAT are a good surrogate of protection against severe disease is fairly speculative and this should be acknowledged repeatedly.

3) Would it be possible to show the inconsistency of the alternative model against long-term data rather than state this without showing the evidence?

4) The authors now do a nice job of explaining the justification for modeling "immunity level" in the body of the paper. Incidentally, I found the response to reviewer #1 to be somewhat circular and confusing. The authors seem to be saying the bi-exponential decay model is based on known B and T cell memory decay profiles but that the model is agnostic about immune mechanisms. Would it not be more accurate to say the profile of immune decay derived from waning vaccine efficacy is an emergent property of the model and that interestingly, is rather consistent with known B cell decay profiles? In any event, the model is explained clearly in the paper itself so it is ok in the end.

REVIEWER COMMENTS

Reviewer #1 (Remarks to the Author):

The authors have provided a reasonable response to the comments I made regarding their manuscript, particularly with regards to my previous comments 2, 3, 4, 5, 6, 7, 9 and 10. However, I still have two concerns with regards to my previous comments 1 and 8.

1. The authors did not explicitly indicate in their rebuttal letters which part of the main text (e.g., p.XX, line XX-XX) corresponds to our comments, although I may have missed their response. In their reply to my previous comment 1, the authors explained the differing assumptions between their model and the Khoury model; however, I could not find a thorough discussion of this in the main text. I believe it would be beneficial to add this discussion to prevent confusion or misunderstandings among non-modelers.

For a general (non-modelling) audience, we feel that this is already covered in paragraphs 3 and 4 of the introduction and in paragraph 5 of the discussion in which we consider the issue about inferring IL from effectiveness data (which is the main difference between our model and the Khoury et al. approach). However, we have now added further text to the methods section to make it clear that the additional modification is in the functional form of the decay profile over time, which is sufficiently flexible to fit both a single and biphasic decay. This is in paragraph 3 of the methods.

2. I am still wondering about the fixed value of "1.61" in my previous comment 8. In order to validate the strong assumption that 1.61 may not be essential to reach the same conclusion, the authors should conduct a sensitivity analysis with different values and report the results in the Supplementary Information at a minimum.

With the recent publication of a paper (Khoury et al. Nature Medicine 2023) that reviews all the variant-adapted vaccines, we have switched to using the central estimate (which by coincidence happens to be 1.61) as this also provides a 95% confidence interval which we can use to sample uncertainty. This has now been propagated through the results and therefore captured in the vaccine effectiveness estimates in Table 3. We have applied this to the estimates for the Moderna ancestral vaccine but have also provided comparative estimates for this applied to the ancestral Oxford/Astrazeneca and Pfizer-BioNTech vaccines in new tables in the supplementary material.

For the associated Figure 3, we have used the central estimate for the scaling, but have now also included new estimates of the scaling against homologous and heterologous strains.

Throughout the text, to be consistent with the paper we are citing, we have modified the language to refer to ancestral vaccines and variant-modified vaccines as the latter contain a range of vaccines, not all of which are bivalent.

Reviewer #2 (Remarks to the Author):

The authors have provided a thorough revision addressing all comments. I recommend that this work is published in Nature Communications.

Reviewer #3 (Remarks to the Author):

The authors have done a great job overall of responding to prior critiques in the paper and in particular acknowledging why the model is not likely to be predictive. The revised discussion is fantastic. I have a few other minor thoughts.

1) The authors state in their response that the purpose of the paper is not to model the immune evasion properties of new variants. This is a fair point but in this sense the model can never be predictive because subsequent waves have a high likelihood of being due to variants whose immune properties are unknown and whose match against existing vaccines is also unknown. It therefore seems necessary to eliminate the terms "prediction" and "predictive" from the paper in favor of "project" and "forecast". Overall, the authors have done a nice job of softening language and providing caveats to the model's ability to project vaccine efficacy over the long term.

We appreciate the sentiment and have checked through the text and made sure that we avoid the word "prediction" throughout.

2) The assumption that NAT are a good surrogate of protection against severe disease is fairly speculative and this should be acknowledged repeatedly.

We have added to the sentence in the discussion to re-iterate this point:

"Furthermore, it does not allow us to gain any further mechanistic insight into the underlying immune dynamics driving the observed vaccine effectiveness against the different clinical endpoints. In particular, we are unable to determine in our analysis whether any single immune marker can be considered a correlate of protection against both mild and severe infection."

3) Would it be possible to show the inconsistency of the alternative model against long-term data rather than state this without showing the evidence?

These were shown in Figure S2 but for clarity we have now included the projections of vaccine effectiveness for this model in the supplementary file (Table S4) as well as for the additive boosting model (Table S5).

4) The authors now do a nice job of explaining the justification for modeling "immunity level" in the body of the paper. Incidentally, I found the response to reviewer #1 to be somewhat circular and confusing. The authors seem to be saying the bi-exponential decay model is based on known B and T cell memory decay profiles but that the model is agnostic about immune mechanisms. Would it not be more accurate to say the profile of immune decay derived from waning vaccine efficacy is an emergent property of the model and that interestingly, is rather consistent with known B cell decay profiles? In any event, the model is explained clearly in the paper itself so it is ok in the end.

Thank you for your comment – yes, on reflection, whilst we were originally motivated by immune system dynamics, the functional form that we fitted was sufficiently flexible and "an emergent property" is a nice way to say this! We have not made any further changes to the text since we have explained the methods in detail.